# Learning Discrete Energy-based Models via Auxiliary-variable Local Exploration

**Hanjun Dai, Rishabh Singh, Bo Dai, Charles Sutton, Dale Schuurmans**
Google Research, Brain Team
{hadai, rising, bodai, charlessutton, schuurmans}@google.com

## Abstract

Discrete structures play an important role in applications like program language modeling and software engineering. Current approaches to predicting complex structures typically consider autoregressive models for their tractability, with some sacrifice in flexibility. Energy-based models (EBMs) on the other hand offer a more flexible and thus more powerful approach to modeling such distributions, but require partition function estimation. In this paper we propose ALOE, a new algorithm for learning conditional and unconditional EBMs for discrete structured data, where parameter gradients are estimated using a learned sampler that mimics local search. We show that the energy function and sampler can be trained efficiently via a new variational form of power iteration, achieving a better trade-off between flexibility and tractability. Experimentally, we show that learning local search leads to significant improvements in challenging application domains. Most notably, we present an energy model guided fuzzer for software testing that achieves comparable performance to well engineered fuzzing engines like libfuzzer.

## 1 Introduction

Many real-world applications involve prediction of discrete structured data, such as syntax trees for natural language processing [1, 2], sequences of source code tokens for program synthesis [3], and structured test inputs for software testing [4]. A common approach for modeling a distribution over structured data is the autoregressive model. Although any distribution can be factorized in such a way, the parameter sharing used in neural autoregressive models can restrict their flexibility. Intuitively, a standard way to perform inference with autoregressive models has a single pass with a predetermined order, which forces commitment to early decisions that cannot subsequently be rectified. Energy-based models [5] (EBMs), on the other hand, define the distribution with an *unnormalized* energy function, which allows greater flexibility by not committing to *any* inference order. In principle, this allows more flexible model parameterizations such as bi-directional LSTMs, tree LSTMs [1, 2], and graph neural networks [6, 7] to be used to capture non-local dependencies.

Unfortunately, the flexibility of EBMs exacerbates the difficulties of learning and inference, since the partition function is typically intractable. EBM learning algorithms therefore employ approximate strategies such as contrastive learning, where positive samples are drawn from data and negative samples obtained from an alternative sampler [8]. Contrastive divergence [9–12], pseudo-likelihood [13] and score matching [14] are all examples of such a strategy. However, such approaches use hand-designed negative samplers, which can be overly restrictive in practice, thus [8, 15, 16] consider joint training of a flexible negative sampler along with the energy function, achieving significant improvements in model quality. These recent techniques are not directly applicable to discrete structured data however, since they exploit gradients over the data space. In addition, the parameter gradient involves an intractable sum, which also poses a well-known challenge for stochastic estimation [17–24].

In this work, we propose *Auxiliary-variable LOcal Exploration (ALOE)*, a new method for discrete EBM training with a learned negative sampler. Inspired by viewing MCMC as a local search in

continuous space, we parameterize the learned sampler using local discrete search; that is, the sampler first generates an initial negative structure using a tractable model, such as an autoregressive model, then repeatedly makes local changes to the structure. This provides a learnable negative sampler that still depends globally on the sequence. As there are no demonstrations for intermediate steps in the local search, we treat it as an auxiliary variable model. To learn this negative sampler, instead of the primal-dual form of MLE [25, 8], we propose a new variational objective that uses *finite-step* MCMC sampling for the gradient estimator, resulting in an efficient method. The procedure alternates between updating the energy function and improving the dual sampler by power iteration, which can be understood as generalization of persistent contrastive divergence (PCD [10]).

We experimentally evaluated the approach on both synthetic and real-world tasks. For a program synthesis problem, we observe significant accuracy improvements over the baseline methods. More notably, for a software testing task, a fuzz test guided by an EBM achieves comparable performance to a well-engineered fuzzing engine on several open source software projects.

## 2 Preliminaries

**Energy-based Models:** Let $x \in \mathcal{S}$ be a discrete structured datum in the space $\mathcal{S}$. We are interested in learning an energy function $f : \mathcal{S} \to \mathbb{R}$ that characterizes the distribution on $\mathcal{S}$. Depending on the space, $f$ can be realized as an LSTM [26] for sequence data, a tree LSTM [1] for tree structures, or a graph neural network [6] for graphs. The probability density function is defined as

$$p_f(x) = \exp\left(f(x) - \log Z_f\right) \propto \exp\left(f(x)\right), \tag{1}$$

where $Z_f := \sum_{x \in \mathcal{S}} \exp\left(f(x)\right)$ is the partition function.

It is natural to extend the above model for conditional distributions. Let $z \in \mathcal{Z}$ be an arbitrary datum in the space $\mathcal{Z}$. Then a conditional model is given by the density

$$p_f(x|z) = \frac{\exp\left(f(x,z)\right)}{Z_{f,z}}, \text{ where } Z_{f,z} = \sum_{x \in \mathcal{S}} \exp\left(f(x,z)\right). \tag{2}$$

Typically $\mathcal{S}$ is a combinatorial set, which makes the partition function $Z_f$ or $Z_{f,z}$ intractable to calculate. This makes both learning and inference difficult.

**Primal-Dual view of MLE:** Let $\mathcal{D} = \{x_i\}_{i=1}^{|\mathcal{D}|}$ be a sample obtained from some unknown distribution over $\mathcal{S}$. We consider maximizing the log likelihood of $\mathcal{D}$ under model $p_f$:

$$\max_f \ \ell\left(f\right) := \mathbb{E}_{x \sim \mathcal{D}}\left[f(x)\right] - \log Z_f. \tag{3}$$

Directly maximizing this objective is not feasible due to the intractable log partition term. Previous work [15, 8] reformulates the MLE by exploiting the Fenchel duality of the log-partition function, *i.e.*, $\log Z_f = \max_q \mathbb{E}_{x \sim q}\left[f(x)\right] - H(q)$, where $H(q) = -\mathbb{E}_q\left[\log q\right]$ is the entropy of $q\left(\cdot\right)$, which leads to a primal-dual view of the MLE:

$$\max_f \min_q \ \bar{\ell}(f,q) := \underbrace{\mathbb{E}_{x \sim \mathcal{D}}\left[f(x)\right]}_{\text{positive sampling}} - \underbrace{\mathbb{E}_{x \sim q}\left[f(x)\right]}_{\text{negative sampling}} - H(q) \tag{4}$$

Although the primal-dual view introduces an extra dual distribution $q\left(x\right)$ for negative sampling, this provides an opportunity to use a trainable deep neural network to capture the intrinsic data manifold, which can lead to a better negative sampler. In [8], a family of flexible negative samplers was introduced, which combines learnable components with dynamics-based MCMC samplers, *e.g.*, Hamiltonian Monte Carlo (HMC) [27] and stochastic gradient Langevin dynamics (SGLD) [28], to obtain significant practical improvements in continuous data modeling. However, the success of this approach relied on the differentiability of $q$ and $f$ over a continuous domain, requiring guidance not only from $\nabla_x f\left(x\right)$, but also from gradient back-propagation through samples, *i.e.*, $\nabla_\phi \bar{\ell}\left(f,q\right) = -\nabla_\phi \mathbb{E}_{x \sim q_\phi}\left[\nabla_x f\left(x\right) \nabla_\phi x\right]$ where $\phi$ denotes the parameters of the dual distribution. Unfortunately, for discrete data, learning a dual distribution for negative sampling is difficult. Therefore this approach is not directly translatable to discrete EBMs.

## 3 Auxiliary-variable Local Exploration

To extend the above approach to discrete domains, we first introduce a variational form of power iteration (Section 3.1) combined with local search (Section 3.2). We present the method for an unconditional EBM, but the extension to a conditional EBM is straightforward.

### 3.1 MLE via Variational Gradient Approximation

For discrete data, learning the dual sampler in the min-max form of MLE (4) is notoriously difficult, usually leading to inefficient gradient estimation [17–24]. Instead we consider an alternative optimization that has the same solution but is computationally preferable:

$$\max_{f,q} \tilde{\ell}(f,q) := \max_f \max_{q \in \mathcal{K}} \mathbb{E}_{x \sim \mathcal{D}}[f(x)] - \mathbb{E}_{x \sim q}[f(x)], \tag{5}$$

$$\mathcal{K} := \left\{ q \ \Big| \ \int q(x) k_f(x'|x) \, dx = q(x'), \forall x' \in \mathcal{S} \right\}, \tag{6}$$

where $k_f(x'|x)$ is any ergodic MCMC kernel whose stationary distribution is $p_f$.

**Theorem 1** *Let* $(f^*, q^*) = \mathrm{argmax}_{f,q} \tilde{\ell}(f,q)$. *If the kernel* $k_f(x'|x)$ *is ergodic with stationary distribution* $p_f$, *then* $f^* = \mathrm{argmax} \, \ell(f)$ *is the MLE and* $q^* = p_{f^*}$.

**Proof** By the ergodicity of $k_f(x'|x)$, there is unique feasible solution satisifying the constraint $\int q(x) k_f(x'|x) \, dx = q(x')$, which is $p_f(x)$. Substituting this into the gradient of $\tilde{\ell}$ yields

$$\mathbb{E}_{x \sim \mathcal{D}}[\nabla_f f(x)] - \mathbb{E}_{x \sim q_f}[\nabla_f f(x)] = 0,$$

verifying that $f$ is the optimizer of (3). ∎

Solving the optimization (5) is still nontrivial, as the constraints are in the function space. We therefore propose an alternating update based on the variational form (5):

- **Update** $q$ **by power iteration:** Noticing that the constraint actually seeks an eigenfunction of $k_f(x'|x)$, we can apply power iteration to find the optimal $q$. Conceptually, this power iteration executes $q_{t+1}(x') = \int q_t(x) k_f(x'|x) \, dx$ until convergence. However, since the integral is intractable, we instead apply a variational formulation to minimize

$$q_{t+1} = \mathrm{argmin}_q D_{KL} \left( \int q_t(x) k_f(x'|x) \, dx \Big|\Big| q \right) = \mathrm{argmin}_q \mathbb{E}_{q_t(x) k_f(x'|x)}[\log q(x')]. \tag{7}$$

In practice, this only requires a few power iteration steps. Also we do not need to worry about differentiability with respect to $x$, as (7) needs to be differentiated only with respect to the parameters of $q$. We will show in the next section that this framework actually allows a much more flexible $q$ than autoregressive, such as a local search algorithm.

- **Update** $f$ **with MLE gradient:** Denote $q_f^* = \mathrm{argmax}_{q \in \mathcal{K}} \tilde{\ell}(f,q)$. Then $q_f^*$ converges to $p_f$. Recall the unbiased gradient estimator for MLE $\ell(f)$ w.r.t. $f$ is

$$\nabla_f \ell(f) = \mathbb{E}_{x \sim \mathcal{D}}[\nabla_f f(x)] - \mathbb{E}_{x \sim q_f^*}[\nabla_f f(x)],$$

By alternating these two updates, we obtain the ALOE framework illustrated in Algorithm 1.

**Connection to PCD:** When we set the number of power iteration steps to be 1, the variational form of MLE optimization can be understood as a function generalized version of Persistent Contrastive Divergence (PCD) [10], where we distill the past MCMC samples into the sampler $q$ [29]. Intuitively, since $f$ is optimized by gradient descent, the energy models between adjacent stochastic gradient iterations should still be close, and the power iteration will converge very fast.

**Connection to wake-sleep algorithm:** ALOE is also closely related to the "wake-sleep" algorithm [30] introduced for learning Helmholtz machines [31]. The "sleep" phase learns the recognition network with objective $D_{KL}(p_f||q)$, requiring samples from the current model. However it is hard to obtain such samples for general EBMs, so we exploit power iteration in a variational form.

### 3.2 Negative sampler as local search with auxiliary variables

Ideally the sampler $q*$ should converge to the stationary distribution $p_f$, which requires a sufficiently flexible distribution. One possible choice for a discrete structure sampler is an autoregressive model, like RobustFill for generating program trees [3] or GGNN for generating graphs [32]. However, these have limited flexibility due to parameters being shared at each decision step, which is needed to handle variable sized structures. Also the "one-pass" inference according to a predefined order makes the initial decisions too important in the entire sampling procedure.

| **Algorithm 1** Main algorithm of ALOE | **Algorithm 2** Update sampler $q$ |
|---|---|
| 1: Input: Observations $\mathcal{D} = \{x_i\}_{i=1}^{|\mathcal{D}|}$ <br> 2: Initialize score function $f$, sampler $q$. <br> 3: **for** $x \sim \mathcal{D}$ **do** <br> 4:     Sample $(\hat{x}, \tilde{x})$ from $q(\hat{x})k_f(\tilde{x}\vert\hat{x})$ <br> 5:     Update $f$ with $-\nabla_f f(x) + \nabla_f f(\tilde{x})$ <br> 6:     Update $q$ using Algorithm 2 <br> 7: **end for** | 1: Input: Current model $f$ <br> 2: **for** $i \leftarrow 1$ to # power iteration steps **do** <br> 3:     Sample $\tilde{x}$ from $q$, and get $x$ from $k_f(\cdot\vert\tilde{x})$. <br> 4:     Sample trajectories $\{\mathbf{x}_{0:t^j}^j\}_{j=1}^N$ for $x$ using <br>      Eq (13) or Eq (14). <br> 5:     Update $q$ with gradient from Eq (12). <br> 6: **end for** |

Figure 1: ALOE for learning unconditional discrete EBMs. Algorithms are similar for conditional case. We demonstrate with a single example, but in practice batched optimization is used.

Intuitively, humans do not generate structures sequentially, but perform successive refinement. Recent approaches for continuous EBMs have found that using HMC or SGLD provides more effective learning [8, 12] by exploiting gradient information. For discrete data, an analogy to gradient based search is local search. In discrete local search, an initial solution can be obtained using a simple algorithm, then local modification can be made to successively improve the structure.

By parameterizing $q$ as a local search algorithm, we obtain a strictly more flexible sampler than the autoregressive counterpart. Specifically, we first generate an initial sample $x_0 \sim q_0$, where $q_0$ can be an autoregressive distribution with parameter sharing, or even a fully factorized distribution. Next we obtain a new sample using an editor $q_A(x_i\vert x_{i-1})$, where $q_A(\cdot\vert\cdot) : \mathcal{S} \times \mathcal{S} \mapsto \mathbb{R}$ defines a transition probability. We also maintain a stop policy $q_{\text{stop}}(\cdot) : \mathcal{S} \mapsto [0, 1]$ that decides when to stop editing. The overall local search procedure yields a chain of $\mathbf{x}_{0:t} := \{x_0, x_1, \dots, x_t\}$, with probability

$$q(\mathbf{x}_{0:t}; \phi) = q_0(x_0) \prod_{i=1}^{t} q_A(x_i\vert x_{i-1}) \prod_{i=0}^{t-1}(1 - q_{\text{stop}}(x_i))q_{\text{stop}}(x_t) \tag{8}$$

where $\phi$ denotes the parameters in $q_0$, $q_A$ and $q_{\text{stop}}$. The marginal probability of a sample $x$ is:

$$q(x; \phi) = \sum_{t, \mathbf{x}_{0:t}:t \leq T} q(\mathbf{x}_{0:t}; \phi)\mathbb{I}\left[x_t = x\right], \text{ where } T \text{ is a maximum length,} \tag{9}$$

which we then use as the variational distribution in Eq (7). The variational distribution $q$ can be viewed as a latent-variable model, where $x_0, \dots, x_{t-1}$ are the latent variables. This choice is expressive, but it brings the difficulty of optimizing (7) due to the intractability of marginalization. Fortunately, we have the following theorem for an unbiased gradient estimator:

**Theorem 2** *Steinhardt and Liang [33]: the gradient with respect to parameters $\phi$ has the form*

$$\nabla_\phi \log q(x; \phi) = \mathbb{E}_{q(\mathbf{x}_{0:t}\vert x_t=x;\phi)}\left[\nabla_\phi \log q([\mathbf{x}_{0:t-1}, x]; \phi)\right] \tag{10}$$

*where* $q(\mathbf{x}_{0:t}\vert x_t = x; \phi) \propto q(\mathbf{x}_{0:t}; \phi)\mathbb{I}\left[x_t = x\right]$.

In above equation, $q(\mathbf{x}_{0:t}\vert x_t = x; \phi)$ is the posterior distribution given the final state $x$ of the local search trajectory, which is hard to directly sample from. The common strategy of optimizing the variational lower bound of likelihood would require policy gradient [34] and introduce extra samplers. Instead, inspired by Steinhardt and Liang [33], we use importance sampling with self-normalization to estimate the gradient in (10). Specifically, let $s_x(\mathbf{x}_{0:t-1})$ be the proposal distribution of the local search trajectory. We then have

$$\nabla_\phi \log q(x; \phi) = \mathbb{E}_{s_x(\mathbf{x}_{0:t-1})}\left[\frac{q(\mathbf{x}_{0:t}\vert x_t = x; \phi)}{s_x(\mathbf{x}_{0:t-1})}\nabla_\phi \log q([\mathbf{x}_{0:t-1}, x]; \phi)\right] \tag{11}$$

In practice, we draw $N$ trajectories from the proposal distribution, and approximate the normalization constant in $q(\mathbf{x}_{0:t}\vert x_t = x; \phi)$ via self-normalization. The Monte Carlo gradient estimator is:

$$\begin{aligned}
\nabla_\phi \log q(x; \phi) &\simeq \frac{1}{N}\sum_{j=1}^{N}\frac{q(\mathbf{x}_{0:t^j}^j\vert x_{t^j} = x; \phi)}{s_x(\mathbf{x}_{0:t^j-1}^j)}\nabla_\phi \log q([\mathbf{x}_{0:t^j-1}^j, x]; \phi) \\
&\simeq \frac{1}{N}\sum_{j=1}^{N}\frac{q(\mathbf{x}_{0:t^j}^j; \phi)}{s_x(\mathbf{x}_{0:t^j-1}^j)\sum_{k=1}^{N}q(\mathbf{x}_{0:t^k}^k; \phi)}\nabla_\phi \log q_\phi([\mathbf{x}_{0:t^j-1}^j, x]; \phi)
\end{aligned} \tag{12}$$

The self-normalization trick above is also equivalent to re-using the same proposal samples from $s_x(\mathbf{x}_{0:t-1})$ to estimate the normalization term in the posterior $q(\mathbf{x}_{0:t}\vert x_t = x; \phi)$. Then, given

a sample $x$, a good proposal distribution for trajectories needs to guarantee that every proposal trajectory ends exactly at $x$. Below we propose two designs for such proposal.

**Inverse proposal:** Instead of randomly sampling a trajectory and hoping it arrives exactly at $x$, we can walk backwards from $x$, sampling $x_{t-1}, x_{t-2}, \ldots, x_0$. We call this an inverse proposal. In this case, we first sample a trajectory length $t$. Then for each backward step, we sample $x_k \sim A'(x_k | x_{k+1})$. For simplicity, we sample $t$ from a truncated geometric distribution, and choose $A'(\cdot|\cdot)$ from the same distribution family as the forward editor $q_A(\cdot|\cdot)$, except that $A'$ is not trained. In this case we have

$$s_x(\mathbf{x}_{0:t-1}) = \text{Geo}(t) \prod_{i=0}^{t-1} A'(x_i | x_{i+1}) \tag{13}$$

Empirically we have found that the learned local search sampler will adapt to the energy model with a different expected number of edits, even though the proposal is not learned.

**Edit distance proposal:** In cases when we have a good $q_0$, we design the proposal distribution based on shortest edit distance. Specifically, we first sample $x_0 \sim q_0$. Then, given $x_0$ and the target $x$, we sample the trajectory $\mathbf{x}_{1:t-1}$ that would transform $x_0$ to $x$ with the minimum number of edits. For the space of discrete data $\mathcal{S} = \{0, 1\}^d$, the number of edits equal the hamming distance between $x_0$ and $x$; if $S$ corresponds to programs, then this corresponds to the shortest edit distance. Thus

$$s_x(\mathbf{x}_{0:t-1}) \propto q_0(x_0) \mathbb{I}\left[t = \text{ShortestEditDistance}(x_0, x)\right] \tag{14}$$

Note that such proposal only has support on shortest paths, which would give a biased gradient in learning the local search sampler. In practice, we found such proposal works well. If necessary, this bias can be removed: For learning the EBM, we care only about the distribution over end states, and we have the freedom to design the local search editor, so we could limit it to generate only shortest paths, and the edit distance proposal would give unbiased gradient estimator.

**Parameterization of $q_A$:** We restrict the editor $q_A(\cdot|x_{i-1})$ to make local modifications, since local search has empirically strong performance [35]. Also such transitions resemble Gibbs sampling, which introduces a good inductive bias for optimizing the variational form of power iteration. Two example parameterizations for the local editor are:

- If $x \in \{0, 1, \ldots, K\}^d$, then $q_A(\cdot|x_{i-1}) = \text{Multi}(d) \times \text{Multi}(K)$, where the first multinomial distribution decides a position to change, and the second one chooses a value for that position.
- If $x$ is a program, the editor chooses a statement in the program and replaces with a generated statement. The statement selector follows a multinomial with arbitrary dimensionality using the pointer mechanism [36], while the statement generater can be an autoregressive tree generator.

The learning algorithm for the sampler and the overall learning framework is summarized in Figure 1. Please also refer to our open sourced implementation for more details [1].

## 4 Related work

**Learning EBMs:** Significant progress has recently been made in learning *continuous* EBMs [12, 37], thanks to efficient MCMC algorithms with gradient guidance [27, 28]. Interestingly, by reformulating contrastive learning as a minimax problem [38, 8, 16], in addition to the model [15] these methods also learn a sampler that can generate realistic data [39]. However learning the sampler and gradient based MCMC require the existence of the gradient with respect to data points, which is unavailable for discrete data. These methods can also be adapted to discrete data using policy gradient, but might be unstable during optimization. Also for continuous data, Xie et al. [29] proposed an MCMC teaching framework that shares a similar principle to our variational power method, when the number of power iterations is limited to 1 (Algorithm 2). Our work is different in that we propose a local search sampler and novel importance proposal that is more suitable for discrete spaces of structures.

For *discrete* EBMs, classical methods like CD, PCD or wake-sleep are applicable, but with drawbacks (see Section 3.1). Other recent work with discrete data includes learning MRFs with a variational upper bound [40], using the Gumbel-Softmax trick [41], or using a residual-energy model [42, 43] with a pretrained proposal for noise contrastive estimation [44], but these are not necessarily suitable for general EBMs. SPEN [45, 46] proposes a continuous relaxation combined with a max-margin principle [47], which works well for structured prediction, but could suffer from mode collapse.

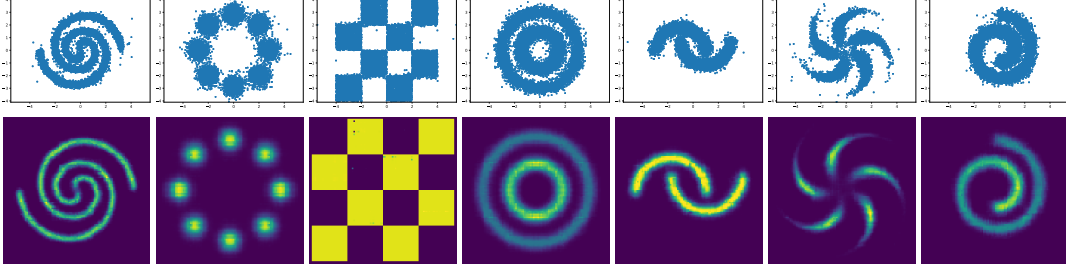

Figure 2: Visualization of learned energy model and sampler. From left to right: 2spirals, 8gaussians, checkerboard, circles, moons, pinwheel, swissroll. Due to the limited space, please refer to Figure A.1 in appendix for the visualization of training samples.

**Learning to search:** Our parameterization of the negative sampler with auxiliary-variable local search is also related to work on learning to search. Most work in that literature considers learning the search strategy given demonstrations [48–51]. When no supervised trajectories are available, policy gradient with variance reduction is typically used to improve the search policy, in domains like machine translation [52] and combinatorial optimization [35]. Our variational form for power iteration circumvents the need for REINFORCE, and thereby gains significant stability in practice.

**Other discrete models:** There are many other models for discrete data, like invertible flows for sequences [53, 54] or graphs [55, 56]. Recently there is also interest in learning non-autoregressive models for NLP [57–59]. The main focus of ALOE is to provide a new learning algorithm for EBMs. Comparing EBMs and other discrete models will be interesting for future investigation.

# 5 Experiments

## 5.1 Synthetic problems

We first focus on learning unconditional discrete EBMs $p(x) \propto \exp f(x)$ from data with an unknown distribution, where the data consists of bit vectors $x \in \{0, 1\}^{32}$.

**Baselines:** We compare against a hand designed sampler and a learned sampler from the recent literature. The hand designed sampler baseline is PCD [10] using a replay buffer and random restart tricks [12], which has shown superior results in image generation. The learned sampler baseline is the discrete version of ADE [8]. Please refer to Appendix A.1 for more details about the baseline setup.

**Experiment setup:** This experiment is designed to allow both a quantitative and 2D visual evaluation. We first collect synthetic 2D data in a continuous space [60], where the 2D data $\hat{x} \in \mathbb{R}^2$ is sampled from some unknown distribution $\hat{p}$. For a given $\hat{x}$, we convert the floating-point number representation (with precision up to $1e^{-4}$) of each dimension into a 16-bit Gray code.[2] This means the unknown true distribution in discrete space is $p(x) = \hat{p}([\texttt{GrayToFloat}(x_{0:15})/1e^4, \texttt{GrayToFloat}(x_{16:31})/1e^4])$. This task is challenging even in the original 2D space, compounded by the nonlinear Gray code. All the methods learn the same score function $f$, which is parameterized by a 4-layer MLP with ELU [61] activations and hidden layer size= 256. ADE and ALOE learns the same form of $q_0$. Since the dimension is fixed to 32, $q_0$ is an autoregressive model with no parameter sharing across 32 steps. For ALOE we also use Gibbs sampling as the base MCMC sampler, but we only perform one pass over 32 dimensions, which is only $1/10$ of what PCD used.

**Main results:** To quantitatively evaluate different methods, we use MMD [62] with linear kernel (which corresponds to $32 - \text{HammingDistance}$) to evaluate the empirical distribution between true samples and samples from the learned energy function. To obtain samples from $f$, we run $20 \times 32$ steps of Gibbs sampling and collect 4000 samples. We can see from Table 1 that ALOE consistently outperforms alternatives across all datasets. ADE variant is worse than PCD on some datasets, as REINFORCE based approaches typically requires careful treatment of the gradient variance.

We also use VAE [63] or autoregressive model to learn the discrete distribution, where the results are shown in the "Other" section of Table 1. Note that these models are different, so the numerical

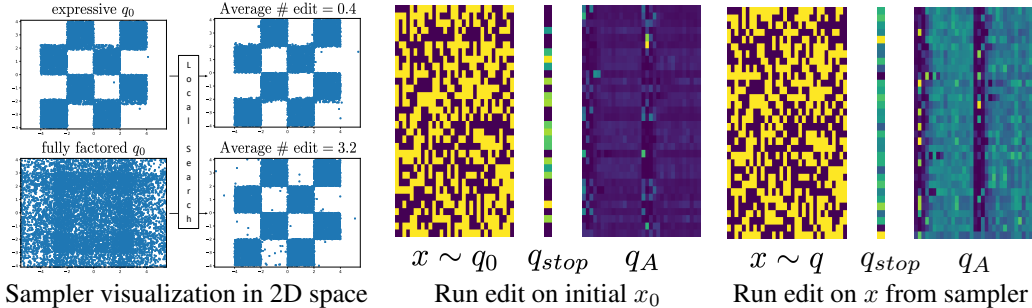

| expressive $q_0$ | Average # edit = 0.4 | | $x \sim q_0$ | $q_{stop}$ | $q_A$ | $x \sim q$ | $q_{stop}$ | $q_A$ |
| fully factored $q_0$ | Average # edit = 3.2 | | | | | | | |

Sampler visualization in 2D space | Run edit on initial $x_0$ | Run edit on $x$ from sampler

Figure 3: Visualization of learned local search sampler in 2D (left) and original discrete Gray code (mid + right) space. See Section 5.1 for more information.

Table 1: Synthetic results with MMD-hamming ($\times \mathbf{1e^{-3}}$) as evaluation metric, and the lower the better. * denote the discrete adaptation of its original method for continuous domain.

|  | | 2spirals | 8gaussians | circles | moons | pinwheel | swissroll | checkerboard |
|---|---|---|---|---|---|---|---|---|
| | PCD-10* [10, 12] | 34.73 | 0.3 | -0.3 | 0.48 | -0.42 | -0.49 | -1.04 |
| | ADE* [8] | 33.4 | -0.28 | 2.01 | 2.16 | 7.64 | 6.12 | -0.69 |
| | ALOE | **30.37** | **-0.97** | **-0.83** | **-0.64** | **-0.64** | **-0.58** | **-1.7** |
| Ablation | ADE-fac | 236.6 | 65.7 | 261.7 | 248.6 | 187.2 | 95.3 | 78.2 |
| | ALOE-fac-noEdit | 51.24 | 91.2 | 5.97 | 76.8 | 59.7 | 15 | 2.98 |
| | ALOE-fac-edit | 32.6 | 3 | **-1.5** | 1.27 | 5.02 | 0.44 | **-2.03** |
| Other | AutoRegressive | 32.7 | -0.3 | -0.8 | -0.45 | **-1.27** | 0.31 | -0.2 |
| | VAE | 35.2 | 2.09 | 0.16 | 1.1 | 0.85 | 2.05 | -0.77 |

comparison is mainly for the sake of completeness. ALOE mainly focuses on learning a given energy based model, rather than proposing a new probabilistic model.

**Visualization:** We first visualize the learned score function and sampler in Figure 2. To plot the heat map, we first uniformly sample 10k 2D points with each dimension in $[-4, 4]$. Then we convert the floating-point numbers to Gray code to evaluate and visualize the score under learned $f$. Please refer to Appendix A.1 for more visualizations about baselines. In Figure 3, we visualize the learned local search sampler in both discrete space and 2D space by decoding the Gray code. We can see ALOE gets reasonable quality even with a weak $q_0(x) = \prod_{i=1}^{32} q_0(x[i])$, and it automatically adapts the refinement steps according to the quality of $q_0$.

**Gradient variance:** Here we empirically justify the necessity of the variational power iteration objective design in (7) against the REINFORCE objective. We train ADE and ALOE (with only $q_0$ for comparison) on `pinwheel` data, and plot the negative log-likelihood of EBM (estimated via importance sampling) and the Monte Carlo estimation of gradient variance

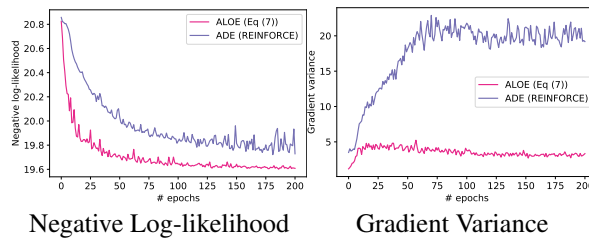

Negative Log-likelihood | Gradient Variance

Figure 4: Training objective and gradient variance.

in Figure 4. We can clearly see ALOE enjoys lower variance and thus faster and better convergence than REINFORCE based methods for EBMs.

**Ablation:** Here we try to justify the necessity of both local edits and the variational power iteration objective. **a)**To justify the local edits, we use a fully factorized initial $q_0$, and compare ALOE-fac-noEdit (no further edits) against ALOE-fac-edit (with $\leq16$ edits). ALOE-fac-edit performs much better than the noEdit version. We use a weak $q_0$ here since we don't need many edits when $q_0$ is the powerful MLP with no parameter sharing (which is not feasible in realistic tasks). Nevertheless, ALOE automatically learns to adapt number of edits as studied in Figure 3 left. **b)**We also show the objective in (7) achieves better results than the REINFORCE objective from ADE.

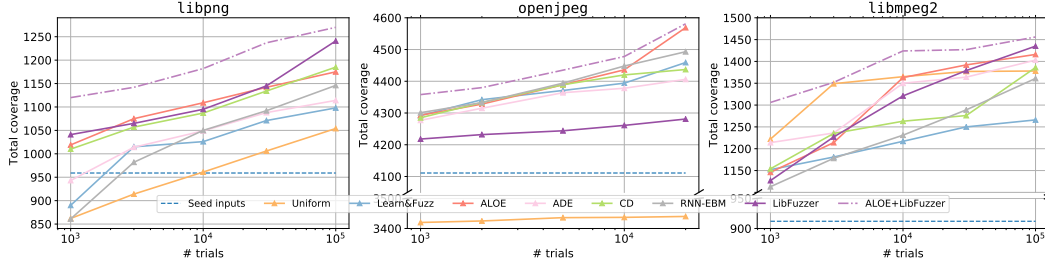

Figure 5: Coverage statistics on different softwares with different amount of test inputs generated.

## 5.2 Generative fuzzing

A critical step in software quality assurance is to generate random inputs to test the software for vulnerabilities, also known as fuzzing [64]. Recently learning based approaches have shown promising results for fuzzing [4, 65]. In this section we focus on the generative fuzzing task, where we first learn a generative model from existing seed inputs (a set of software-dependent binary files) and then generate new inputs from the model to test the software.

**Experiment setup:** We collect three software binaries (namely `libpng`, `libmpeg2` and `openjpeg`) from OSS-Fuzz[3] as test target. For all ML based methods, we use the seed inputs that come with OSS-Fuzz to learn the generative model. As each test input for software can be very large (*e.g.*, a media file for `libmpeg2`), we train a truncated EBM with a window size of 64. Specifically, we learn a conditional EBM $f(x|y)$, where $x \in \{0, \ldots, 255\}^{64}$ is a chunk of byte data and $y \in \{0, 1, \ldots\}$ is the position of this chunk in its original file.

During inference, instead of generating test inputs from scratch (which would be too difficult to generate 1M bytes while still being parsable by the target software), we use the learned model to modify the seed inputs instead. To modify $i$-th byte of the byte string $\mathbf{x}$ using learned EBM, we sample the byte $b \propto \exp(f([\mathbf{x}_{i-31}, \ldots, b, \ldots, \mathbf{x}_{i+32}]|i))$ by conditioning on its surrounding context.

We compare against the following generative model based methods:

- `Learn&Fuzz` [4]: this method learns an autoregressive model from sequences of byte data. We adapt its open-source implementation[4]. To use the autoregressive model for mutating the seed inputs, we perform the edit by sampling $x_i \sim p(\cdot|\mathbf{x}_{0:i-1})$ conditioned on its prefix.
- `ADE` [8]: This method parameterizes the model and initial sampler $q_0$ in the same way as ALOE.
- `CD`: As PCD is not directly applicable for conditional EBM learning, we use CD instead.
- `RNN-EBM`: It treats the autoregressive model learned by `Learn&Fuzz` as an EBM, and mutates the seed inputs in the same way as other EBM based mutations.

We also use uniform sampling (denoted as `Uniform`) over byte modifications as a baseline, and include `LibFuzzer` coverage with the same seed inputs as reference. Note that `LibFuzzer` is a well engineered system used commercially for fuzzing, which gathers feedback from the test program by monitoring which branches are taken during execution. Therefore, this is supposed to be superior to generative approaches like EBMs, which do not incorporate this feedback.

For all methods, we generate up to 100k inputs with 100 modifications for each. The main evaluation measure is *coverage*, which measures how many of the lines of code, branches, and so on, are exercised by the test inputs; higher is better. This statistic is reported by `LibFuzzer`[5].

**Results** are shown in Figure 5. Overall the discrete EBM learned by ALOE consistently outperforms the autoregressive model. Suprisingly, the coverage obtained by ALOE is comparable or even better than `LibFuzzer` on some targets, despite the fact that `LibFuzzer` has access to more information about the program execution. In the long run, we believe that this additional information will allow `LibFuzzer` to perform the best, it is still appealing that ALOE has high sample efficiency initially. Regarding several EBM based methods, we can see CD is comparable on `libpng` but for large target like `openjpeg` it performs much worse. ADE performs good initially on some targets but gets worse in the long run. Our hypothesis is that it is due to the lack of diversity, which suggests

Table 2: Program synthesis accuracy on RobustFill tasks [3].

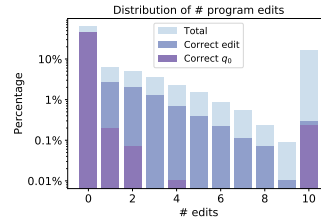

|  | Top-1 Beam-1 | Top-1 Beam-10 | Top-10 Beam-10 |
|---|---|---|---|
| seq2seq-init | 45.86 | 55.49 | 58.66 |
| seq2seq-tune | 47.86 | 57.52 | 60.62 |
| ALOE | **53.57** | **61.99** | **65.29** |

a potential mode drop problem that is common in REINFORCE based approaches. The uniform baseline performs worst in most cases, except on `libmpeg2` early stage. Our hypothesis is that the uniform fuzzer quickly triggers many branches that raise formatting errors, which explains its high coverage initially.

We also combine the test inputs generated by `LibFuzzer` and ALOE (the orange dotted curve, for which the $x$-axis shows the number of samples from each method). The coverage of this combined set of inputs is better than either individually, showing that the methods are complementary.

### 5.3 Program Synthesis

In program synthesis, the task is to predict the source code of a program given a few input-output (IO) pairs that specify its behavior. We evaluate ALOE on the RobustFill task [3] of generating string transformations.The purpose here is to evaluate the effect of proposed local edits in our sampler, so the other methods like ADE or PCD are not applicable here. For full details, see Appendix A.2.

**Experiment setup:** Data is generated synthetically, following Devlin et al. [3]. Each example in the data set is a synthesis task where the input is four IO pairs, the target is a program, and a further six IO pairs are held out for evaluation. The training data is generated on the fly, while we keep 10k test examples for evaluation. Each target program consists of at most 10 sub-expressions in a domain-specific languages which includes string concatenation, substring operations, etc. We report accuracy, which measures when the predicted program is consistent with all 10 IO pairs.

For ALOE we learn a conditional sampler $q(x|z)$ where $x$ is the program syntax tree, and $z$ is the list of input-output pairs. We compare with 3-layer seq2seq model for program prediction. Both seq2seq and ALOE share the same IO-pair encoder. As mentioned in Section 3.2, the initial distribution $q_0$ is the same as seq2seq autoregressive model, while subsequent modifications $A(x_{i+1}|x_i)$ adds/deletes/replaces one of the subexpressions. We train baseline seq2seq with 483M examples (denoted as seq2seq-init), and fine-tune with additional 264M samples (denoted as seq2seq-tune) with reduced learning rate. ALOE initializes $q_0$ from seq2seq-init and set it to be fixed, and train the editor $q_A(\cdot|\cdot)$ with same additional number of samples with the shortest edit importance proposal (14).

**Results:** We report the top-$k$ accuracy with different beam-search sizes in Table 2. We can see ALOE outperforms the seq2seq baseline by a large margin. Although the initial sampler $q_0$ is the same as seq2seq-init, the editor $q_A(\cdot|\cdot)$ is able to further locate and correct sub-expressions of the initial prediction. In the figure to the right of Table 2, We also visualize the number of edits our sampler makes on the test set. In most cases $q_0$ already produces correct results, and the sampler correctly learns to stop at step 0. From 1 to 9 edits we can see the editor indeed improved from $q_0$ by a large margin. There are many cases which require 10 or more edits, in which case we truncate the local search steps to 10. Some of them are difficult cases where the sampler learns to ask for more steps, while for others the sampler keeps modifying to semantically equivalent programs.

## 6 Conclusion

In this paper, we propose ALOE, a new algorithm for learning discrete EBMs for both conditional and unconditional cases. ALOE learns a sampler that is parameterized as a local search algorithm for proposing negative samples in contrastive learning framework. With an efficient importance reweighted gradient estimator, we are able to train both the sampler and the EBM with a variational power iteration principle. Experiments on both synthetic datasets and real-world software testing and program synthesis tasks show that both the learned EBM and local search sampler outperforms the autoregressive alternative. Future work includes better approximation of learning local search algorithms, as well as extending it to other discrete domains like chemical engineering and NLP.

## Broader Impact

We hope our new algorithm ALOE for learning discrete EBMs can be useful for different domains with discrete structures, and it furthers the general research efforts in this direction of generative models of discrete structures. In this paper, we present its application to program synthesis and software fuzzing. A positive outcome of improved performance in program synthesis would be that it can help democratize the task of programming by allowing people to express their desired intent using input-output examples without the need of learning complex programming languages. Similarly, a positive outcome of improvements in software fuzzing could allow software developers to identify bugs and vulnerabilities quicker and in turn improve software reliability and robustness.

A possible negative outcome could be that malicious attackers might also use such technology to discover software vulnerabilities and use it for undesirable purposes [66]. However, this outcome is not specific to our technique but more generally applicable to the large research field of software fuzzing, and there is a large amount of work in the fuzzing field for accounting ethical considerations. For example, the vulnerabilities typically found by fuzzers is first responsibly disclosed to corresponding software teams [67] that gives them enough time to patch the vulnerabilities before the bugs and vulnerabilities are released publicly.

## Acknowledgments and Disclosure of Funding

We would like to thank Sherry Yang for helping with fuzzing experiments. We would also like to thank Adams Wei Yu, George Tucker, Yingtao Tian and anonymous reviewers for valuable comments and suggestions.

## Footnotes

[1]https://github.com/google-research/google-research/tree/master/aloe

[2]https://en.wikipedia.org/wiki/Gray_code

[3]https://github.com/google/oss-fuzz

[4]https://github.com/google/clusterfuzz/tree/master/src/python/bot/fuzzers/ml/rnn

[5]https://llvm.org/docs/LibFuzzer.html

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
