[Supplementary Material]

# Appendix

## A  More experiment details

### A.1  Synthetic experiments

**Baseline configuration**   We adapt two most recent techniques for learning EBMs into discrete case, namely Du and Mordatch [1] and Dai et al. [2]. Specifically:

- **PCD based**:  Du and Mordatch [1] extends the PCD method with replay buffer and random restart. We adapt these tricks in learning discrete EBMs. Specifically, we use Gibbs sampling for $K \times 32$-steps as the MCMC sampler, where $K$ is set to 10. Instead of always inheriting from previous MCMC samples, we tune the restart rate in $\{0.05, 0.1, 1\}$.
- **ADE based**: ADE solves the same minimax problem in Eq (4), but instead directly minimizes the objective $L(q) := -\mathbb{E}_{x \sim q}\left[f(x)\right] - H(q)$. To make ADE work in discrete case, optimizing $L(q)$ requires the policy gradient technique with variance reduction [3–6], where the gradient estimator becomes $\nabla_q L(q) = \mathbb{E}_{x \sim q} \nabla \log q(x)(-f(x) - \log q(x) - 1)$. This also resembles the learning of GAN [7] on sequences [8] or graphs [9], except the additional entropy regularization term and constant. We use A2C [3] to learn ADE for discrete EBMs. As ADE uses alternating minimization for minimax problem, we tune the learning rate ratio and synchronization frequency between energy function and sampler learning in $\{0.2, 0.5, 1\}$ and $\{1:1, 1:3, 1:5\}$, respectively.

Figure A.1: 2D visualization of samples from the ground truth distribution.

Figure A.2: visualization of learned discrete EBMs using different methods.

**More visualizations**   In Figure A.1 we also visualize the samples obtained from the ground truth distribution and visualize them in 2D space.  Compared to Figure 2 we can see our learned sampler can almost perfectly recover the true distribution.  The checkerboard seems to be the most difficult one among these datasets, as for both PCD and ADE baselines the learned model is much worse than the one learned by ALOE. We find that in this case

the distribution is not smooth as it has sharp boundaries for each "square" in the distribution. Thus below we study how the learned sampler behaves for ADE algorithm in this case. In Figure 3 in main paper we have studied ALOE with different design choices of $q_0$, where a weak $q_0$ like fully factored distribution can still get reasonable results. Instead in Figure A.3 we can see that, for ADE, different parameterizations of the sampler will make quite different behaviors. The MLP sampler is an autoregressive one with non-sharing parameters, while the RNN sampler has the shared parameters across different steps. This clearly shows the limitation of autoregressive model with parameter sharing, and also

$q_0 =$MLP          $q_0 =$RNN

Figure A.3: ADE with different samplers.

the necessity of learning sampler with local search to improve the weak initial sampler $q_0$.

**Implementation details**   Here we provide more details on the instantiation of ALOE on the synthetic tasks. Below we first cover the parameterization details.

The energy function is an MLP with dimensions of $[32, 256, 256, 256, 1]$, where 32 is the input size, and 256 is the hidden layer size. We use ELU as the activation function.

For ADE and ALOE, the $q0$ is parameterized with either autoregressive model or a factorized model. For the factorized model, we simply learn 32-dimensional vector that represents the logits of each dimension independently. For the autoregressive model, there can be two choices. The first one uses LSTM (which we denote as `RnnSampler`) to encode the bits, where LSTM has hidden size of 256 and 1 layer. All the dimensions share the same predictor that predicts the binary bit from the latent embedding obtained by LSTM. The predictor is an MLP with size $[256, 512, 2]$ with ELU activation. Another alternative is to use MLP to encode the bits, as we know the maximum length is 32 beforehand (which is not practical in general). This way we encode the history using 31 MLPs, where the $i$-th MLP has size $[i, 512, 512, 256]$ that embeds the prefix of length $i$, and use the shared predictor to predict the bit at current position.

ALOE has additional components, which are editor $q_A(\cdot|\cdot)$ and stop policy $q_{\text{stop}}$. The editor only needs to predict the location for modification, as once the location is given one can simply flip that bit. It is parameterized into $[32, 512, 512, 32]$ with ELU as activation function and softmax at the end. The stop policy is parameterized by an MLP with layers $[32, 512, 512, 1]$ with ELU activation and sigmoid in the last output.

We use the `Inverse proposal` where $A'(\cdot|\cdot)$ is a uniform distribution that samples a random location for modification. To avoid sampling the same position twice, we first permute the locations and then pick the first $k$ locations as the proposal trajectory, where $k$ is the number of edits that is sampled from a geometric distribution, with the truncation at 16.

## A.2   Program synthesis experiments

**Grammar:** We use the following grammar for RobustFill programs.

| | | |
|---|---|---|
| $\langle program \rangle$ | $\rightarrow$ | $\langle ExprList \rangle$ |
| $\langle ExprList \rangle$ | $\rightarrow$ | $\langle expr \rangle \mid \langle expr \rangle \langle ExprList \rangle$ |
| $\langle expr \rangle$ | $\rightarrow$ | 'ConstStr' $\langle ConstExpr \rangle \mid$ 'SubStr' $\langle SubstrExpr \rangle$ |
| $\langle ConstExpr \rangle$ | $\rightarrow$ | ']' $\mid$ ',' $\mid$ '-' $\mid$ '.' $\mid$ '@' $\mid$ ''' $\mid$ '"' $\mid$ '(' $\mid$ ')' $\mid$ ':' $\mid$ '%' |
| $\langle SubstrExpr \rangle$ | $\rightarrow$ | $\langle Pos \rangle \langle Pos \rangle$ |
| $\langle Pos \rangle$ | $\rightarrow$ | $\langle ConstPos \rangle \mid \langle RegPos \rangle$ |
| $\langle ConstPos \rangle$ | $\rightarrow$ | -4 $\mid$ -3 $\mid$ -2 $\mid$ -1 $\mid$ 0 $\mid$ 1 $\mid$ 2 $\mid$ 3 $\mid$ 4 |
| $\langle RegPos \rangle$ | $\rightarrow$ | $\langle ConstTok \rangle \mid \langle RegexTok \rangle$ |
| $\langle ConstTok \rangle$ | $\rightarrow$ | $\langle ConstExpr \rangle \langle p2 \rangle \langle direct \rangle$ |
| $\langle RegexTok \rangle$ | $\rightarrow$ | $\langle RegexStr \rangle \langle p2 \rangle \langle direct \rangle$ |
| $\langle p2 \rangle$ | $\rightarrow$ | $\langle ConstPos \rangle$ |
| $\langle direct \rangle$ | $\rightarrow$ | 'Start' $\mid$ 'End' |

86 ⟨*RegexStr*⟩ → `[A-Z]([a-z])+'|`[A-Z]+'|`[a-z]+'|`\d+'|`[a-zA-Z]+'|`[a-zA-Z0-9]+'
87 |`\s+'|`^'|`$'

**Data generator:** We use following configurations for generating synthetic data for program synthesis:

- The maximum number of types of tokens in input strings is set to 5.
- The maximum length of input strings is 20.
- The maximum length of output strings is 50.
- The total number of input-output examples per synthesis task is 10.
- The number of public input-output example pairs is 4.
- The number of private input-output example pairs is 6.

The learned synthesizer uses the 4 public IO pairs for synthesize the program, and evaluate against all 10 IO pairs. It is considered correct if it is consistent with these 10 IO pairs.

**Parameterization:** We use a 3-layer LSTM with hidden size of 256 to encode each input and output sequences, respectively. Then each IO pair is represented by concatenating the sequence embeddings of input and output strings. The set of inputs is obtained by max-pooling over the IO-pair embeddings, which will be served as the context for program synthesis.

For $q_0$ we use a 3-layer LSTM with hidden size of 256 for predicting program tokens. For ALOE we parameterize the $q_A$ with two components: the position predictor $q_{\text{pos}}$ and the modified expression $q_{\text{expr}}$. $q_{\text{pos}}$ embeds the current program using 3-layer bidirectional LSTM, and predict the position using pointer mechanism [10]. Note that the selected position must be the start or end of an existing $< expr >$ in above grammar, which indicates whether we want to modify or insert a new $< expr >$ in this position. $q_{\text{expr}}$ predicts the new expression using another 3-layer LSTM, and is allowed to make empty prediction (which corresponds to delete an expression in current program). As the program heavily relies on the context free grammar to make it valid, we utilize the technique in grammarVAE [11] to mask out invalid production rules during program generation.

## A.3 Fuzzing experiment

| Software | # seed files | file size (bytes) | # training samples for ALOE |
|---|---|---|---|
| libpng | 170 | 104 - 12,901 | 146,507 |
| openjpeg | 36 | 233 - 7,885,684 | 27,572,688 |
| libmpeg2 | 131 | 10,581 - 50,000 | 6,119,237 |

Table A.1: Data statistics for generative fuzzing experiments. We use window size 64 for ALOE to obtain chunks of data from the raw byte streams.

**Data statistics:** We test different approaches against three target softwares. The OSS-Fuzz project comes with different set of seed inputs for different target softwares. These inputs are served as training samples for both ALOE and Godefroid et al. [12], and will be used as seed inputs for libFuzzer as well. Table A.1 displays the data statistics. Note that ALOE trains a conditional EBM with chunked data from the original raw byte streams, in order to handle huge files. We use chunk size 64 by default. Thus for a file with size $L$ where $L \geq 64$, there will be $L - 64 + 1$ training samples for ALOE.

**Parameterization:** We use a three-layer MLP to parameterize the energy function, where for the input layer, we use embedding size equals to 4 for the byte string. For the negative sampler, we parameterize $q_0$ with LSTM. $q_A$ consists of two parts, namely $q_{\text{pos}}$ which predicts which position to modify using an MLP, and $q_{\text{value}}$ which predicts a new value for that position using another MLP. We use Eq (13) for training such EBM.