[Reviews · NeurIPS 2020]

Review 1

Summary and Contributions: This paper proposes a new strategy for learning conditional and unconditonal energy based models and demonstrate its benefits on synthetic and real data. The paper looks very solid to me, but I have honestly no idea what it is about - this is not a problem of the paper, just very poor selection of me as a reviewer! I gave it one pass to at least provide a minimal review

Strengths: The paper looks very solid to me, the text reads well and everything feels legit. The broader impact statement is probably the best one in the 6 papers I got to review!

Weaknesses: I did not like the sentence: "We present the method for an unconditional EBM, but the extension to a conditional EBM is straightforward." From how easy it is to go from unconditional to conditional in the previous paragraph I kind of believe it, but I would prefer if you add this straight forward extension to the supplementary material

Correctness: I can't judge on this

Clarity: Yes, it reads nicely and seems clear

Relation to Prior Work: for my understanding yes, I however don't know the surrounding literature

Reproducibility: Yes

Additional Feedback:


Review 2

Summary and Contributions: The paper proposes an approach to learning energy-based models (EBMs) over discrete variables. In particular, the authors propose an approach to jointly learning a sampler, which is intended to produce samples approximately from the model distribution, along with the model parameters. This sampler is used to approximate the gradient of the negative log likelihood wrt model parameters. The paper contributes an approach to learning a sampler, as well experimental results on several tasks, and comparisons to baselines.

Strengths: - The idea of learning a sampler in the way the paper proposes is very interesting and appears promising. - Learning discrete EBMs is an important and timely topic. - The paper considers an interesting application domain, namely, software fuzzing. - The authors obtain good empirical results.

Weaknesses: - Some of the claims made by the authors seem imprecise (see below), and the presentation could be more clear/streamlined. - Parts of the proposed approach seem somewhat ad-hoc, and would benefit from better empirical or theoretical motivation (see below). - Some of the baselines in the experiments seem weak or missing.

Correctness: Some concerns: - As I understand it the paper makes two major methodological proposals: that we learn a sampler by minimizing (7), and that this sampler should produce samples through local, stochastic edits. I would expect an ablation showing whether these choices are both necessary, but I don't think there is one. - If I understand correctly, the edit-distance proposal distribution doesn't have support over trajectories that are too long, whereas the true posterior does, and so the importance-sampling estimate will be biased. - For the Learn&Fuzz autoregressive baseline in the generative fuzzing experiments, just sampling the edit from p(x_i | x_{0:i-1}) (line 251) seems unnecessarily weak: why not condition on the entire context (including subsequent bytes, by running the model forward on the whole byte string for each value of b) just as in the EBM case? - I can't tell whether the results in Table 1 use held-out samples from the true distribution or the training ones; I would expect some evaluation of generalization. - Also, there don't seem to be EBM-baselines for the final two experimental applications. It might not be necessary to include every baseline in experiments, especially if one generally performs poorly, but can we conclude just from the experiments in Table 1 that the baseline EBMs won't work well on the fuzzing and synthesis tasks?

Clarity: I think the paper could be presented more clearly and in a more streamlined way: much of Section 2 (including the discussion of primal-dual view of MLE and Theorem 1) seem unnecessary for the subsequent development. In particular, as far as I can tell, the authors are simply interested in a new way of obtaining samples from the model distribution to allow for estimating the gradient of the negative log likelihood, which is a fairly standard approach to learning EBMs. Somewhat minor: the discussion of the drawbacks of autoregressive models (lines 20-22) seems somewhat imprecise: it's not the case that inference over autoregressive models needs to be done in a way such that earlier decisions can't be rectified, nor that autoregressive models imply a particular inference order (though it might be more tractable to use a particular order). Additional minor comments: - Line 66: should be + H(q) - Equations 9 & 10: it might be clearer if the marginal q was distinguished symbolically from the the per-action q's.

Relation to Prior Work: The discussion of relation to prior work is informative.

Reproducibility: Yes

Additional Feedback: --- Update after rebuttal: thanks for your responses and the updated experiments and results! I think the paper certainly seems stronger experimentally now, and I'm increasing my score. However, after reading the rebuttal I still wonder whether a simple autoregressive q distribution (e.g., given by by an RNN or Transformer) with no editing might perform well; this approach would not require any importance sampling-based gradient estimation. I also encourage the authors to significantly simplify their presentation. My understanding is the authors have presented an approach and loss function for learning a sampler that can sample from the model distribution, which allows us to get estimates of the log likelihood gradient. I would encourage the authors to just say this, and leave out the primal-dual formulations, power iteration, variational formulations, etc, which I think obscure rather than clarify.


Review 3

Summary and Contributions: [After rebuttal/discussion: Thank you for the detailed rebuttal, and in particular running some of the additional experiments that were raised. However, after the discussion period (and another re-read of the paper), I have decided to downgrade my score. I think the presentation could be revamped substantially to make the proposed approach much more understandable. As it stands the mathiness seems to encumber rather than help the exposition (and given the approximations/relaxations, it is not clear that the proposed approach is so grounded in theory). There were also several issues with baselines (e.g. raised by R2) that were only partially addressed in the rebuttal.] The authors propose an approach to learn discrete energy-based models whose distribution is given by normalizing the energy. MLE with such models is in general intractable due to the partition function. In order to make the problem tractable, the authors first consider a primal-dual view of MLE via a variational formulation of the log partition, which introduces a variational distribution q. Then they approximate the variational distribution q with a learned sampler. Learning for q is performed via weighted importance sampling instead of the usual policy gradient. The proposed approach is tested on toy datasets and generative fuzzing.

Strengths: - While autoregressive models are tractable and work well even in domains where there is no natural ordering (e.g. graphs), investigating undirected (energy-based) distributions over such combinatorial sets is an important open problem. - The use of variational formulations and estimating q with a learned distribution, combined with various tricks (inverse proposal, edit distance proposal), is creative and sound.

Weaknesses: - The results seem somewhat preliminary and only tested on toy-ish domains. It would have been interesting to apply this approach on domains such as machine translation (though program synthesis experiments are interesting!) - I would have liked to have seen this method tested against tractable density estimation methods on the toy dataset as well, such an autoregressive model that assumes some ordering, given that with enough data, an autoregressive model may work well enough. - As I understand it, the log likelihood evaluation of this approach is still intractable, which means that one has to come up with various heuristic objectives to evaluate the learned model. - For some of the tricks (e.g. edit distance proposal), there are many domains in which this would be nontrivial to calculate or wouldn't make sense (e.g. edit distance on graphs). This limits the generality of the proposed approach.

Correctness: Yes

Clarity: Yes

Relation to Prior Work: Yes

Reproducibility: Yes

Additional Feedback:


Review 4

Summary and Contributions: This paper introduces a new maximum likelihood estimator, ALOE, for learning discrete structured energy models. It uses an alternative form of the primal-dual formulation of the MLE where we alternate between learning the dual distribution and learning the model by updating its parameters. The dual distribution q is parameterised to consist of an initial distribution, followed by a set of transition moves resembling a local search. This approach enables learning models that give impressive results synthetic tasks and tasks from the program synthesis literature.

Strengths: The work is very clearly written and builds on existing literature in a really nice way. The algorithm is justified by previous theoretical work into energy models. It is a novel approach in not relying as much on continuous relaxations but really leaning into a search-based approach. The evaluation is convincing and highly relevant baselines are used. This work is relevant to the NeurIPS community as it builds on ideas like contrastive divergence, energy models, variational inference, and learning to search, all topics that have been well-represented in the community for many years.

Weaknesses: The work sometimes feels like an algorithm was found to empirically to work and theory was found to justify it later as much of the approach feels more complicated than it might need to be. Great pains are taken in the paper to explain why a simpler approach isn't taken as often the objectives being proposed are difficult to optimise. I would have appreciated many another experiment exploring the variance of the MLE as REINFORCE is used for learning q and it would useful to know how the method compares with a more straightforward VAE architecture for learning discrete models.

Correctness: The claims and method seem reasonable and the empirical method seems well thought out.

Clarity: The paper is well-written and it's mostly straightforward for follow the reasoning of the paper.

Relation to Prior Work: All relevant literature seems to be mentioned at least in passing.

Reproducibility: Yes

Additional Feedback:

[Author Response · NeurIPS 2020]

We thank the reviewers for the comments, which we will incorporate into the next version. For brevity we denote the
reviewers by [**R1**][**R2**][**R3**][**R4**]. We have included additional baselines and ablations in Table 1 (synthetic) and Figure
1 (fuzzing) (described more below). Overall ALOE still performs consistently comparable or better than alternatives.

[**R1**] **Conditional EBM:** This extension re-
quires changes only to the parameterizations
of energy function, samplers (into $q(x; z)$) with-
out affecting the overall framework. We will
elaborate more in our revision.
[**R2**] **ablation on minimizing (7) and local ed-**
**its:** Thanks for the suggestions. We found both

Table 1: Ablations for ALOE; compared to Table 1 in main paper.

| Methods | 2sprs | 8gauss | cir | moon | pwhl | sroll | ckbd |
|---|---|---|---|---|---|---|---|
| ALOE | **30.37** | **-0.97** | -0.83 | **-0.64** | -0.64 | **-0.58** | -1.7 |
| ADE-fac | 236.6 | 65.7 | 261.7 | 248.6 | 187.2 | 95.3 | 78.2 |
| ALOE-fac-noEdit | 51.24 | 91.2 | 5.97 | 76.8 | 59.7 | 15 | 2.98 |
| ALOE-fac-edit | 32.6 | 3 | **-1.5** | 1.27 | 5.02 | 0.44 | **-2.03** |
| AutoRegressive | 32.7 | -0.3 | -0.8 | -0.45 | **-1.27** | 0.31 | -0.2 |
| VAE | 35.2 | 2.09 | 0.16 | 1.1 | 0.85 | 2.05 | -0.77 |

were separately helpful through ablations. **a)**To justify the local edits, we use a fully factorized initial $q_0$, and compare
ALOE-fac-noEdit (no further edits) against ALOE-fac-edit (with $\leq 16$ edits). ALOE-fac-edit performs much better
than the noEdit version. We use a weak $q_0$ here since we don't need many edits when $q_0$ is the powerful MLP with no
parameter sharing (which is not feasible in realistic tasks). ALOE automatically learns to adapt number of edits, as
studied in Fig 3 (left) and Table 2 (right) in main paper. **b)**We also show (7) achieves better results than the REINFORCE
objective from ADE [ref 8 in paper], when we compare ADE-fac that uses the same sampler as ALOE-fac-noEdit.

[**R2**] **Table 1 results** All methods are evaluated against the same held-out test set.
[**R2**] **Edit-distance bias:** We agree with the reviewer. Our experiments show that
the bias is not a big issue in practice. If necessary, this bias can be removed: For
learning the EBM, we care only about the distribution over end states, and we have
the freedom to design $q$, so we could limit $q$ to generate only shortest paths.
[**R2**] **Use RNN like EBM for fuzzing:** As suggested, we include RNN-EBM in Fig
1, which uses RNN as score function and is otherwise the same as our setting. It is
indeed better than prefix based sampling, but is still inferior to ALOE in general.
[**R2**] **EBM baselines on other tasks:** For program synthesis we mainly evaluate
the effect of local edits in our sampler, so the other methods are not applicable;
for fuzzing we here include ADE and CD (it is a conditional EBM and PCD's
buffer is not directly applicable). From the results in Fig 1 we can see ALOE still
outperforms baselines consistently. CD is comparable on `libpng` but for large target
like `openjpeg` it performs much worse. ADE performs good initially on some targets
but gets worse in the long run. This is due to the lack of diversity, which suggests
a potential mode drop problem that is common in REINFORCE based approaches.
[**R2**] **"Clarity: Theorem 1 seems unnecessary":** Thanks for your suggestion.
Theorem 1 is needed to motivate the "variational form of power method" in Algorithm
2 and in (7). We will make this more clear in our revision.
[**R2**] **Minor "...drawbacks of autoregressive...imprecise":** Fair point. We agree
that autoregressive models can also be used in a way like EBM during inference, but
EBMs can be more general and thus more powerful. We will appropriately weaken
the claims. Also thanks for suggestions on typos and notations. We will address.

[**R3**] **"...toy-ish domains..."** We emphasize that fuzzing is done on real-world soft-
wares with large sample size (see Table A.1 in appendix), where `libfuzzer` baseline
is used in commercial. We will explore more application domains in the future.
[**R3**][**R4**] **other models on toy data:** The main purpose of synthetic experiment is
to compare different learning methods for the *same* EBM. Nevertheless, we have included autoregressive (with LSTM)
and VAE models (with MLP) in Table 1 as suggested. ALOE still performs the best overall. But note that EBMs and
the VAE/autoregressive ones use different models and sampling methods.

Figure 1: More fuzzing results.

[**R3**] **"...evaluation...heuristic..."** Likelihood is not tracable to
compute in EBMs, while using MMD to measure distribution
discrepancies is a common protocol rather than a random heuristic.
[**R3**] **"...tricks...domain specific"** It is common to serialize the
trees (like we used for program synthesis in the paper) and graphs
(e.g., SMILES language). Edit-distance can also be defined directly
on trees (e.g., `gumtree`) and graphs (`GED`).

Negative Log-likelihood      Gradient Variance

[**R4**] **"... complicated.. variance of REINFORCE"** we have included ablations above to justify our design. Regarding
the variance, we plot the gradient variance and learning objective during training (estimated via importance sampling)
for pinwheel data. We can clearly see ALOE enjoys lower variance than REINFORCE based methods for EBMs.

[Meta-Review · NeurIPS 2020]

We have three competent reviews all of which recommend acceptance. In my opinion the main room for improvement involves empirical evaluation. As exact calculation of the log loss is intractable a downstream task evaluation would be nice. But the paper seems acceptable with the current results.